# Effective Intracerebral Connectivity in Acute Stroke: A TMS–EEG Study

**DOI:** 10.3390/brainsci13020233

**Published:** 2023-01-30

**Authors:** Franca Tecchio, Federica Giambattistelli, Camillo Porcaro, Carlo Cottone, Tuomas P. Mutanen, Vittorio Pizzella, Laura Marzetti, Risto J. Ilmoniemi, Fabrizio Vernieri, Paolo Maria Rossini

**Affiliations:** 1Laboratory of Electrophysiology for Translational Neuroscience (LET’S), Institute for Cognitive Sciences and Technologies (ISTC), National Research Council of Italy (CNR), 00185 Rome, Italy; 2Department of Clinical Neurology, University Campus Bio-Medico, 00128 Rome, Italy; 3Department of Neuroscience and Padova Neuroscience Center (PNC), University of Padova, 35128 Padova, Italy; 4Centre for Human Brain Health, School of Psychology, University of Birmingham, Birmingham B15 2TT, UK; 5Department of Neuroscience and Biomedical Engineering, Aalto University School of Science, 00076 Espoo, Finland; 6BioMag Laboratory, Helsinki University Hospital Medical Imaging Center, Helsinki University Hospital, Helsinki University and Aalto University School of Science, P.O. Box 340, FI-00029 HUS Helsinki, Finland; 7Department of Neuroscience, Imaging and Clinical Sciences, University ‘G. d’Annunzio’ of Chieti-Pescara, 66100 Chieti, Italy; 8Institute for Advanced Biomedical Technologies, University ‘G. d’Annunzio’ of Chieti-Pescara, 66100 Chieti, Italy; 9Laboratory of Brain Connectivity, Department of Neuroscience & Neurorehabilitation, IRCCS San Raffaele-Roma, 00163 Rome, Italy

**Keywords:** TMS–EEG, stroke, acute phase, center-on surround-off, central peripheral excitability, electromyography EMG

## Abstract

Stroke is a major cause of disability because of its motor and cognitive sequelae even when the acute phase of stabilization of vital parameters is overcome. The most important improvements occur in the first 8–12 weeks after stroke, indicating that it is crucial to improve our understanding of the dynamics of phenomena occurring in this time window to prospectively target rehabilitation procedures from the earliest stages after the event. Here, we studied the intracortical excitability properties of delivering transcranial magnetic stimulation (TMS) to the primary motor cortex (M1) of left and right hemispheres in 17 stroke patients who suffered a mono-lateral left hemispheric stroke, excluding pure cortical damage. All patients were studied within 10 days of symptom onset. TMS-evoked potentials (TEPs) were collected via a TMS-compatible electroencephalogram system (TMS–EEG) concurrently with motor-evoked responses (MEPs) induced in the contralateral first dorsal interosseous muscle. Comparison with age-matched healthy volunteers was made by collecting the same bilateral-stimulation data in nine healthy volunteers as controls. Excitability in the acute phase revealed relevant changes in the relationship between left lesioned and contralesionally right hemispheric homologous areas both for TEPs and MEPs. While the paretic hand displayed reduced MEPs compared to the non-paretic hand and to healthy volunteers, TEPs revealed an overexcitable lesioned hemisphere with respect to both healthy volunteers and the contra-lesion side. Our quantitative results advance the understanding of the impairment of intracortical inhibitory networks. The neuronal dysfunction most probably changes the excitatory/inhibitory on-center off-surround organization that supports already acquired learning and reorganization phenomena that support recovery from stroke sequelae.

## 1. Introduction

Brain functionality depends crucially on neuronal connections between cerebral areas. Transcranial magnetic stimulation (TMS) allows non-invasive investigation of the functional connections of the human cerebral cortex, with special sensitivity for the motor cortex. By means of rapidly changing magnetic fields reaching the brain undistorted by extra-cerebral tissues, induced electric currents can depolarize and thereby activate cortical neurons [1,2]. In recent years, we have learned to explore the brain’s effective connectivity by recording electroencephalographic activity (EEG) during TMS and obtaining the corresponding TMS-evoked potentials (TEPs). This is TMS–EEG [3,4]. In fact, the TMS–EEG method allows one to observe directly how the stimulation of the cortex evokes synchronized neuronal activity in connected areas with high spatiotemporal specificity [3,4,5,6].

In stroke patients, functional recovery of the upper limb is one of the most challenging goals and, often, the most frustrating. Considering that neural plasticity, one of the key factors involved in functional recovery after stroke, strongly depends on spared and ancillary neuronal networks, the study of effective connectivity in the acute phase can provide relevant information with prognostic value [7,8,9,10,11,12,13,14]. On this line, we used the EEG in combination with TMS to investigate projections originating from the primary motor cortex (M1) of both the left lesioned and right contra-lesion hemispheres. 

Previous studies of the brain organization in the early phase after stroke demonstrated that one major damage effect is the unbalance of interhemispheric mutual modulation with an excessive inhibitory influence of the unaffected hemisphere on the one where the stroke is located. This conceptual framework originated numerous intervention strategies based on constraint-induced movement therapy [15]. In line with the above pioneering observations, functional-imaging studies targeting M1 using TMS in stroke patients with lesions of mono-lateral M1 or its corticospinal projections demonstrated increased recruitment and abnormally decreased short interval cortical inhibition (SICI) of the right contra-lesion M1 within the first month after infarction [16,17,18,19]. 

Experience in subacute stroke patients [20] showed that abnormally decreased SICI of a contra-lesion M1 can only partially be explained by loss of interhemispheric inhibition (IHI) from the lesioned or non-lesioned hemisphere. In fact, a decreased SICI of the contra-lesion M1 did not result in reduced IHI from the lesioned hemisphere’s M1 [20]. Considering that all patients showed excellent recovery of motor function, decreased SICI of the contralesional M1 may represent an adaptive process supporting recovery [20]. In another study, Liepert and colleagues demonstrated an increase in motor output area size and MEP amplitudes from the lesioned hemisphere, speculating enhanced neuronal excitability in the damaged hemisphere for the target muscles after constraint-induced movement therapy. Other studies suggested a role of contra-lesion hemisphere disinhibition in increasing the amplitude of the MEP evoked by stimulation of that hemisphere [21,22]. 

### Aim

We collected TEPs induced by TMS targeting M1 and concurrent MEPs to assess ‘central’ and ‘peripheral’ counterparts of neuronal circuitry for hand motor control modifications in the acute phase after stroke with the aim of uncovering the pattern of inhibition-excitation balances within and between lesioned and contra-lesion hemispheric homologous areas.

Based on this line of reasoning, we used previously described cerebral recruitments induced by M1 TMS as described by TEP-source analysis in multiple previous investigations (Table 1, for review, see [5,6]).

Mainly interested in the balances between inhibitory and excitatory networks, which are indistinguishable in terms of inverse-problem solutions, we approached the new knowledge on functional connectivity with origin in M1 by performing a population analysis comparing stroke patients and healthy controls through the magnitude of evoked activity, in terms of TEP’s global field power, which, at a certain latency, results from three non-mutually exclusive factors: the number of activated neurons, the intensity of their activation, and/or the level of their synchronization.

## 2. Methods

### 2.1. Subjects

Seventeen stroke patients and nine healthy volunteers were enrolled after they had given written informed consent to the experimental protocol previously approved by the institutional Ethics Committee. Patients and healthy volunteers, matched for age, were instructed to abstain from caffeine, alcohol, and medication and to maintain their regular sleep–wake schedule on the day and night before the experimental session. All subjects were right-handed, as evaluated by the Handedness Questionnaire (0.70 ± 0.08). The exclusion criteria established by international safety standards for TMS were followed [23,24]. All the patients were evaluated using the international standardized clinical scale (NIHSS, National Institute of Health Stroke Scale). 

Inclusion criteria for stroke patients were:-Age 18–90;-NIH stroke scale (NIHSS) range 6–24;-Single ischemic stroke in the middle cerebral artery territory of the left hemisphere within 10 days;-Upper arm paresis (upper arm at least NIHSS > 1).-Exclusion criteria were:-Symptom onset more distant than 10 days;-Associated neurological diseases;-Multiple ischemic strokes;-Previous ischemic or hemorrhagic stroke;-TMS contraindication, according to the recommendations of the International Federation of Clinical Neurophysiology (IFCN, [24]);-Compromised vigilance or severe hemodynamic, neurological, or respiratory conditions;-Poor middle cerebral artery insonation through transcranial Doppler;-Hemodynamic carotid stenosis (it could determine a compensatory dilatation in the distal circulation, with a consequent reduction of basal VMR);-Refusal to sign the informed consent.

It has been demonstrated that there are different levels of cerebral intracortical inhibition according to cortical or sub-cortical ischemic lesions [19]. Furthermore, it is well known that in some stroke patients, MEP cannot be elicited, even at the highest stimulator output intensity [19,25]. To standardize our study population, we excluded patients with pure cortical ischemic lesions, and we enrolled only patients in whom MEPs were evident also in response to the stimulation of the affected hemisphere. 

### 2.2. TMS–EEG Experimental Setup and Protocol

TMS-compatible EEG equipment (BrainAmp 32MRplus, BrainProducts GmbH, Munich, Germany) recorded EEG continuously from 32 scalp sites (Fp1, Fp2, F3, F4, C3, C4, P3, P4, O1, O2, F7, F8, T7, T8, P7, P8, Fz, Cz, Pz, FC1, FC2, CP1, CP2, FC5, FC6, CP5, CP6, TP9, TP10, FT9, FT10, FCz of the 10–20 International System) using electrodes mounted on an elastic cap. The Oz electrode served as the ground to have maximal distance from the stimulating coil; linked mastoids defined the reference potential. To avoid overheating of the electrodes located in the vicinity of the stimulating coil, TMS-compatible Ag/AgCl-coated electrodes were used. Skin/electrode impedance was maintained below 5 kOhm. BrainAmp MRplus allows fine adaptation of the TMS stimulus strength by selecting amplifier sensitivity and operational range to prevent saturation under specific stimulus conditions. The sensitivity of 100 nV/bit (signal range/resolution) and an analog/digital-conversion range of 6553.5 mV (±3276.8 mV) were used. 

TMS was performed over the left M1 during multichannel EEG recording, monitoring the coil position stability along the session with a neuronavigation system (NBS system, SofTaxicOptic, Bologna, Italy) which also assured stable coil orientation with respect to the subject’s head. EMG activity from the left and right First Dorsal Interosseous (FDI) muscles was recorded via surface electrodes in belly tendon montage. Magstim SuperRapid magnetic stimulator was used with a figure-of-eight coil having an outer wing diameter of 7 cm (Magstim Company Limited, Whitland, UK). After the EEG cap was attached, the coil was placed tangentially to the scalp over the C3 site of the 10–20 System with the handle pointing backward and laterally at about 45° angle from the midline. The coil was moved in steps of 5 mm to search for the best coil position to induce maximal MEPs from the right FDI—FDI ‘hot spot’. After having identified the FDI hot spot coil position, the FDI resting motor threshold (RMT) was determined as the lowest stimulus intensity eliciting at least 5 MEPs of 50 microV out of 10 consecutive stimuli [24]. EEG and EMG signals were sampled at 5 kHz after bandpass filtering at 50–1000 Hz for EMG and 0.1–500 Hz for EEG.

To mask coil-generated clicks, white noise was continuously delivered to the participant through earphones. We adjusted the masking volume until the participants reported that the TMS click was no longer audible (always below 90 dB of sound pressure). To ensure wakefulness throughout the recording sessions, subjects were required to keep their eyes open, maintaining their gaze on a fixation point during stimulation periods (2–3 min). About 70 magnetic stimuli were delivered at 120% RMT (supra-threshold stimulation) over left and right M1 regions with an irregular inter-stimulus interval in the range of 4–6 s. 

### 2.3. Data Analysis

#### 2.3.1. MEP and EEG Data

In addition to exploiting the TMS-compatible EEG unit (see details above) that is not saturated by the TMS pulse, the residual TMS–EEG artifacts were suppressed using the established data analysis procedures [3,26].

#### 2.3.2. Data-Processing Pipeline for TMS–EEG Data

The signals acquired from EEG during TMS stimulation can be modeled as follows:$$S\left(i,t\right)=T\left(i,t\right)+\sum_{a}A_{a}\left(i,t\right)+N\left(i,t\right)$$
$S\left(i,t\right)$ is the measured signal, $T\left(i,t\right)$ is the signal of interest, $\sum_{a}A_{a}\left(i,t\right)$ is the sum of the artifacts (indexed by *a*), and $N\left(i,t\right)\;$ is the noise in the i -th channel at time *t*. 

The aim of the pre-processing steps is to reduce the artifact term $\sum_{a}A_{a}\left(i,t\right)$. The steps needed for this purpose are indicated below.

A typical EEG recording during TMS is presented in Figure 1A,B. The data are averaged over the repeated TMS stimuli (*t* = 0 ms). As can be seen, the TMS artifact is several orders of magnitude higher than the brain signal. Via software, we simulate the hardware intervention of the sample-and-hold amplifiers [27] (as our TMS system is not equipped with this) by replacing the 5 ms interval around the TMS stimulus (−3 to 2 ms) with the 5 ms interval of the baseline (−8 to −3 ms) (Figure 1).
Figure 1Example of an EEG recording during TMS. (**A**). Original data averaged over the TMS stimulus (*t* = 0). (**B**). The data after replacing the [−3, 2] ms interval with the 5-ms interval of the baseline [−8, −3] ms.
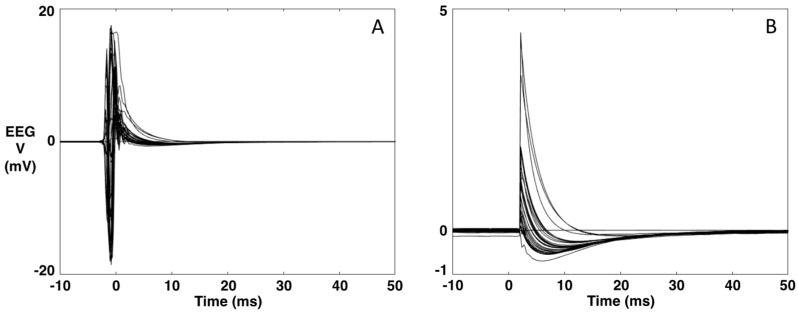

A channel that has been minimally corrupted by a TMS stimulus artifact is chosen through visual inspection. The data are re-referenced against this channel. In our case, the Cz channel is the channel that is least corrupted by the TMS artifact.Application of the SOurce-Utilized Noise Discarding (SOUND) algorithm for cleaning the channels separately for the baseline and the data after the TMS stimulus [26]. TMS–EEG data can contain various other noise sources that corrupt individual channels or cause strange voltage patterns on the scalp. The SOUND algorithm cross-validates EEG channels via consecutive inverse and forward computations. The cross-validation outputs an estimate for the noise distribution across the EEG channels, which is used to form a spatial Wiener filter that highlights the neuronal EEG signals. SOUND filters out those signal components that are not likely to originate from intracranial post-synaptic currents, e.g., electrode-polarization, line-noise, and electrode-movement artifacts [26,28,29].

In our case, the parameters of the SOUND algorithm were set as follows: 5 interactions and a lambda value of 1000 for the post-stimulus and a lambda value of 100 for the baseline. The performance of SOUND is shown in Figure 2.

4.Re-referencing the data to the mean reference. At this point, the data are processed by the SSP–SIR (Signal-Space Projection—Source-Informed Reconstruction) algorithm for automatic cleaning of residual artifacts, such as muscle artifacts [28,30]. SSP–SIR substantially improved the signal quality of artifactual TMS–EEG data, causing minimal distortion in the neuronal signal components. In the SSP–SIR approach, the artifact signal subspace containing TMS-evoked muscle artifacts is estimated from the high-pass-filtered (cutoff frequency 100 Hz) data using principal component analysis. The rationale is that EEG signals above 100 Hz mainly consist of non-neuronal signals. The estimated artifact subspace is projected out using SSP [30]. The remaining artifact-suppressed signals are used to estimate an equivalent source distribution model exploiting anatomical brain constraints. When solving the inverse problem, the artifact dimensions must be projected out also from the lead field (or gain) matrix. In the final SIR step, the obtained source estimates are projected back onto the original signal space.5.Finally, independent component analysis using the fastICA [31] algorithm is applied to the averaged data to eliminate eye blinking and suppress random noise (first and fourth rows of Figure 3). Using data cleaned by SOUND, which is most efficient in eliminating the TMS and the motor artifacts, we used a deep experience in exploiting statistical features of ocular artifacts, thus well identified by the ICA approach [32,33].

#### 2.3.3. Stroke vs. Healthy Volunteer Comparison

Data were analyzed by exploiting the TMS-induced MEP when stimulating, as we did, the M1 area. That is, stroke patients were compared to control subjects in terms of the MEPs as well as TEPs. The evoked EEG comparison was executed between stroke and healthy volunteer populations statistically comparing the Global Field Power (GFP) of data after pre-analyses described above by steps 1 to 5 and calculating the average of the squared signal of each of the 32 channels (in the average reference). 

GFP, introduced by Lehmann and Skrandies [34], is a reference-free quantity, calculated by the numerical procedures which assess the degree of ‘hilliness’ (or ‘relief’, or electrical strength) of the fields. The approach is to consider all possible potential differences in the field (for n electrodes, n*(n−1)) with equal weight and, thus, compute the reference-free, mean potential difference (global field power) at each moment in time using the formula:$$GFP\left(t\right)=\sqrt{\frac{1}{2n}}\overset{n}{\underset{i=1}{\sum }}\overset{n}{\underset{j=1}{\sum }}{\left(u_{i}\left(t\right)+u_{j}\left(t\right)\right)}^{2}$$
where *n* is 32, the number of our recording EEG channels, and *u* is the re-referenced channel value at time *t* (with 0 the TMS stimulus delivery at left or right M1).

Pointwise statistical analysis was performed on the GFP by two-sample permutations *t*-test using 5000 permutations.

The entire data and statistical analyses were performed by Matlab (The Mathworks, Inc., Natick, MA, USA). 

Results were reported for statistical significance *p* < 0.05.

## 3. Results

### 3.1. Enrolled Population

Patients (pts) had a mean age of 74.5 years with a standard deviation (SD) of 7.5 years; healthy controls had a mean age of 72.7 years with an SD of 5.8 years. The TMS resting motor thresholds (RMTs), as a percentage of the stimulator output, were 60%, SD 10% and 60%, SD 9% for the right- and left-hand muscles, respectively. Lesion sites of the left hemisphere were distributed in basal nuclei (2 pts), thalamus (2 pts), frontal-insular (4 pts), frontal-subcortical (2 pts), cortico-subcortical Rolandic (7 pts). The study was conducted, on average, 7 ± 2 days after symptom onset.

### 3.2. MEP Amplitude 

MEP amplitude values were log-normally distributed and were therefore log-transformed as y = log (MEP + 1) to achieve a good approximation to a Gaussian distribution, as assessed using the Shapiro–Wilk test and to limit the potentially detrimental effect of right-skewed outliers. 

In the patient population, the MEP amplitude was significantly different between hemispheres, being lower when stimulating the ischemic (left) hemisphere than when stimulating the contralateral one (Amplitude (log): 2.02 vs. 2.40: *p* < 0.001, Figure 4 left vs. right side light blue traces) or stimulating the corresponding hemisphere in the control group (Amplitude (log): 2.02 vs. 2.24: *p* < 0.001, Figure 4 left side, light blue vs. red traces). However, the MEP amplitude resulting from TMS to the right contra-lesion M1 in stroke patients was larger than the corresponding amplitude in healthy subjects (Amplitude (log): 2.40 vs. 2.22: *p* < 0.001, Figure 4 right side, light blue vs. red traces). No statistical differences in MEP amplitudes were found stimulating the right and left hemisphere in healthy subjects (Amplitude (log): 2.24 vs. 2.22: *p* = 0.522).

### 3.3. Quality of Artefact Removal

The quality of the artifact removal procedures is evident from the TMS-evoked potentials (TEP) shown in Figure 3 where the signal-to-noise- ratio allows clear identification of the typical components of the brain response to M1 TMS. Notably, these components are evident in single subjects’ TEPs when stimulating the lesioned left and contra-lesion hemispheres right M1 in stroke patients. We describe TEPs via the superimposition of all 32 channels (first row), the Global Field Power (GFP, time by time squared potential summed across all channels), and the topographies at the latencies corresponding to maxima of the GFP (red asterisks in channel superimposition). Similarly, one can appreciate the quality of cleaned data in the elderly healthy controls (Figure 3).

### 3.4. Global Intracerebral Effective M1 Connectivity 

We statistically compared the dynamics of activation propagation within the brain via the time evolution of the GFP in response to M1 TMS in the left ipsilesional- and right contralesional-hemisphere. The comparison used a random permutation procedure to assess the significance of the differences. When stimulating the left M1, we found that stroke patients had an enhanced responsiveness (GFP intensity) compared to healthy volunteers, reaching statistical significance around 100 ms. When stimulating the right contra-lesion hemisphere, patients had a reduced responsiveness at about the same latency with respect to healthy volunteers. Right and left M1 stimulation induced similar TEPs in healthy volunteers (Figure 5).

## 4. Discussion

The key result of our study is that, concomitantly with the expected reduction of central-peripheral projections, resulting in a reduced motor evoked potential amplitude in the paretic than non-paretic hand and healthy controls, excessive excitability occurred intra-cortically in the ipsilesion hemisphere, with a reduction in the contralesion hemisphere as indexed by TEP Global Field Power.

TMS–EEG allows to study cortico–cortical interactions [35,36] with better temporal resolution than the TMS–fMRI technique because of its high temporal resolution; it also offers a direct evaluation of neuronal activity, instead of PET studies, which are based on flow/metabolic data. Moreover, it allows, in the case of M1 TMS, the concurrent assessment of brain and muscle characterization.

Lack of glucose and oxygen in neural tissue involved in ischemic stroke leads to progressive neuronal degeneration and necrosis. These events—which also include subsequent axonal degeneration—lead to reduced synaptic activation and altered interaction with contiguous and remote (i.e., contralateral) connected brain areas. That is why it is very important to analyze network connectivity rather than merely investigate the reactivity of a localized region, as recently shown on EEG via graph-theoretical methods [37].

This study evidenced the alteration in neuronal network transmission that occurs after an acute ischemic stroke in the left hemisphere. Previous studies have demonstrated an indirect alteration of cerebral neuronal transmission through modification of MEP amplitude and evidenced, from the very start of the ischemic event, a modification of cerebral excitability also in areas far from the stroke [14,21,38,39,40,41,42,43]. In our study, in agreement with these previous works, MEP amplitudes were significantly different between hemispheres, being lower when stimulating the ischemic hemisphere than the contra-lesional. On the other hand, larger MEPs resulted from stimulation of the contra-lesion M1 in stroke patients than from the stimulation of the same hemisphere in healthy subjects. These findings confirmed hyperexcitability of the contra-lesion hemisphere in subjects with acute stroke [19] as a symptom of reorganization of the brain also in the contra-lesion hemisphere. A great novelty comes from the possibilities opened by the TMS–EEG investigation method, which allows for the evaluation of intracortical communication with origin in the peri-lesion areas, and which revealed a central hyper-excitability in conjunction with the depletion of peripheral projections. The two concomitant phenomena assessed by the simultaneous collection of peripheral and intracortical transmissions when stimulating the central nodes of the cortico-spinal tract suggest an impact of the stroke on the local phenomenon of the on-center off-surround neuronal learning mechanism. Since the increase in total power may result from a greater number of activated neurons, a greater intensity of their activation, and/or a higher level of their synchronization, it is inferred that the effect of stroke is to increase the overall synchronization of the recruited node. Very plausibly, this results from the alteration of the specific on-center off-surround mechanism. The local inhibitory networks are key elements in the acquisition of skilled sensorimotor control both at the level of the motor cortex [44] and of the spinal cord [45], as well as in sensory-perceptual acuity acquisition [46], and it is implemented by surround lateral inhibition processes. Notably, the balance between the activity of homologous hemispheric areas carried out by local inhibitory mechanisms common to fine neuronal network processing ability is, thus, a ubiquitous structural–functional mechanism that supports the plastic adaptation and learning processes of the brain [47,48]. In other words, the interplay between homologous structures in the two hemispheres is a critically integrated part of functional inhibitory–excitatory circuitry that supports the functionality of the body segment they control. Here, an effect of the bilateral circuit alterations in acute stroke comes from the peripheral projection when stimulating the contra-lesion right M1, which evokes a hyper-excitable muscular response. The interplay between homologous areas in the two hemispheres, mediated by excitatory projections from one side to the inhibitory network of the other [49,50,51], contributes to the skill acquisition of behavioral control, also influencing recovery after stroke [52,53]. Here, the central hyper-excitability of local transmission in response to the same stimulus that evokes reduced peripheral projections emerges from the first instants within the first 100 ms. Historical and more recent studies of TMS–EEG TEPs indicates that the component emerging around 100 ms reflects GABA-mediated inhibitory recruitments [54,55,56].

A limitation of our study, which can be overcome in future studies, is the lack of investigating activated regions by solving the inverse problem, allowing us to trace specific recruitments and any differences in activated networks recruited at latencies following M1 activation.

In conclusion, we believe that current advances in understanding the hemispheric effects of unilateral stroke may complement previously observed imbalances between the homologous regions of the two hemispheres with critical information about the circuit mechanisms that encode sensorimotor information. Notions of contralateral hemisphere-induced hyper inhibition of perilesional regions have guided strategies related to Constraint-Induced Movement Therapy in stroke rehabilitation. The indications of the present work, with a hyper-excitability of the affected hemisphere in terms of reduced efficacy of local inhibitory networks, stimulate new strategies, probably to be implemented with appropriate neuromodulation interventions. Interestingly, future studies can also focus on these phenomena using simpler and more widely available experimental settings, starting with the advances obtained with the excellent TMS–EEG technology.

## Figures and Tables

**Figure 2 brainsci-13-00233-f002:**
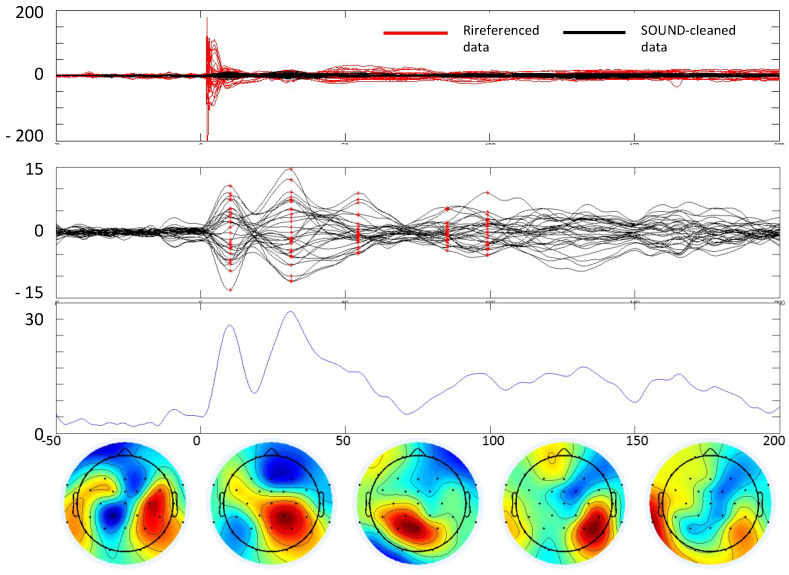
TEP extraction. Example of cleaning the TMS–EEG data of a subject with stroke. In all boxes, *t* = 0 is the TMS stimulus delivery at the scalp. From top to bottom. First box: comparison between data referenced to Oz and data cleaned by the SOUND algorithm. Second box: data cleaned with SSP–SIR and ICA. Third box: Global field power envelope of the cleaned data. *x*-axis time in ms, *y*-axis in μV (first and second boxes), and μV^2^ (third box). Fourth row: Topographies of the cleaned data at 5 time instants where an evoked field is clearly notable (latencies indicated in red in the second box).

**Figure 3 brainsci-13-00233-f003:**
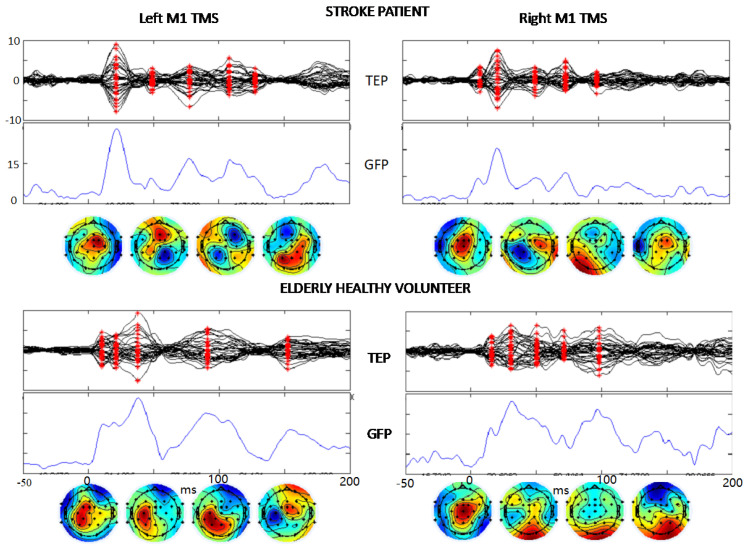
TEP description. The same structure of boxes from second to fourth rows of Figure 2 showing the TMS-evoked EEG potentials in a stroke patient and a healthy volunteer in response to the TMS stimulation of left and right M1 (*t* = 0). The vertical scales are equal in the two hemispheres and both persons, μV for TEP and μV^2^ for GFP.

**Figure 4 brainsci-13-00233-f004:**
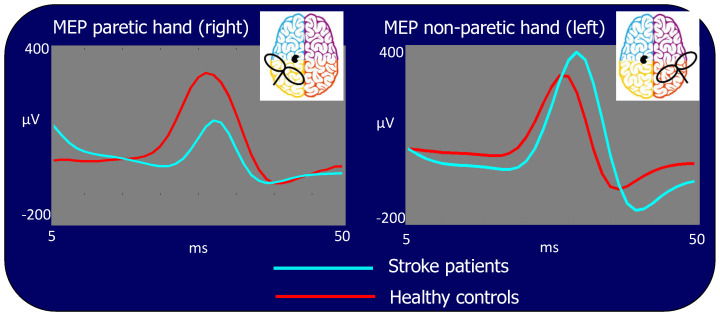
Motor evoked potentials (MEP) for stimulation of the lesioned left and contra-lesion right hemispheres. Time scale refers to *t* = 0 as the TMS stimulus delivery at the scalp. In light blue: grand average of the patient group, in red: grand average of the control group. The MEP is significantly smaller in patients than in control subjects when TMS is performed in the left lesioned and larger than in controls in the right contra-lesion M1s.

**Figure 5 brainsci-13-00233-f005:**
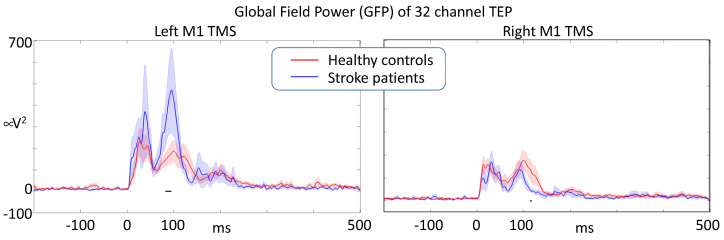
After artifact removal analysis, Global Field Power of the responses induced by single-pulse M1 TMS (*t* = 0) describing the overall effective intracerebral connectivity of M1 projections. Stroke lesion was in the left hemisphere in all patients (black horizontal segments indicate a significant difference at *p* < 0.05).

**Table 1 brainsci-13-00233-t001:** M1-TMS TEP components.

Wave	Topography	Functional Connectivity	Synaptic Substrate
N7	F3	ipsi-lateral motor associative, PM	NMDA
P13	Fp2 F4 F8 C4 T4 T6	contra-lateral homolog M1	GABA_A_
N18	P3	ipsi-lateral PPC	
P30	Extended^C^	contra-lateral thalamo-cortical nodes	
N44	Extended^C&I^	ipsi- and contral-lateral M1	
P60	T5		GABA_B_
N100	Extended^I^	ipsi-lateral M1	

Based on previous knowledge, we indicate for each of the component (wave) typical of the TEP-characterized response to left M1 TMS within the first 100 ms, the prevalent channel/s with the nomenclature of the International EEG system (topography), the hypothesized neuronal recruited areas (functional connectivity) and the main synaptic transmission mediating the projection of fibers connecting the recruited areas (synaptic substrate). PM = premotor area, NMDA = N-methyl-D-aspartate glutamatergic receptor, GABA = gamma-aminobutyric acid including the two classes: the more rapid GABA_A_ and the slower GABA_B_, PPC = post-parietal cortex, Extended^C^ = Fp1 Fp2 Fz F4 F8 Cz T4 Pz P4 T6 O1 O2. Extended^I^ = Fp1 Fp2 Fz F3 F7 Cz T3 P3 T5 Pz O1 O2. Extended^C&I^ = distribution involving all channels with a rostro-caudal gradient.

## Data Availability

TMS-EEG data will be made available upon request.

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
