# Peer review of "Effective Intracerebral Connectivity in Acute Stroke: A TMS–EEG Study"

_brainsci, 2023, doi:10.3390/brainsci13020233_

Round 1
Reviewer 1 Report
I read this study with interest, yet I expect the authors to revise and consolidate the manuscript. My comments and questions are listed below.
Major:
Please add line numbers in the manuscript. It will be convenient for the communication between reviewers and authors
In the introduction session, I expect that previous TMS-EEG studies can be introduced in detail, rather than only mentioning the SICI. a) The changes to the neural networks should be introduced, and b) the mathematic methods used in the Methods session should be introduced.
For Figure 1, the [-3,2] ms interval is replaced by the baseline data from -8 to -3 ms. Since the manipulation is performed within a short time window, please zoom in to a short period, like -10 to 50 ms. It does not make sense to show us the time series from -100 to 500 ms.
Does the GFP signal reflect intracerebral connectivity?
I expect the authors provide more information about the data processing. The SOUND algorithm and the SSP-SIR algorithms have been published, yet it is still necessary to talk about the contents of the algorithms.
How does ICA help to eliminate eye blinking?
I don’t think the discussion covers all the important findings in this paper. For example, what is the findings in the cortico-cortical interactions? It would be necessary to briefly list all the findings in the Results session and consider the neural implications of each MEP and TES
Minor:
Please define Global Field Power as GFP on Page 7.
Author Response
please see the attachment, thank you for your kind comments.

Reviewer 2 Report
Reading the manuscript written by Giambattistelli et al., entitled: Effective Intracerebral Connectivity in Acute Stroke: a TMS–EEG Study was really interesting.
In general, the article is easy to read, quite well designed and can be of interest to readers and researchers. The methodologies are appropriate and aligned with the proposed objectives.
Further details and some suggestions on how to improve your work are described below:
- English is good, but there is spelling, punctuation and some grammar issues.
- Please summarize the main theme of the paper in a graphical abstract.
Author Response

(The authors gave the same response as above.)

Reviewer 3 Report
In this manuscript, the authors used a TMS system and TMS-compatible EEG set to assess TMS-induced evoked potential (TEPs) and motor evoked potential (MEPs) in acute stroke survivors. The TMS pulses were delivered to the primary motor cortex (M1). They found that the ipsilateral hemisphere was overly excited with respect to the contralateral hemisphere, and the similar hemisphere in healthy control subjects.
Please consider the following revision:
1) It is always good to mention the software tool used. For example: Is it EEGLAB or other Matlab-based codes/toolboxes?
2) Figure 1: it is hard to find differences visually in panel B and D. Can the authors perhaps enlarge the X-axis, for example to be [-100,+200] msec.
3) Section 2.3.3: Statistical tests or hypothesis tests shall be mentioned and statistical differences were considered significant if p < 0.05. Also, there is a typo in the Global Field Power, and kindly revise clearly how this variable was derived/measured.
4) Can the authors provide statistical outputs of the analysis in Section 3.5? This is related to Q2 above on how the Global Field Power was calculated and compared.
5) Putting many acronyms reduces readability for the general public. I suggest the authors consider typing out some that repeat in the manuscript, e.g. HV, CLH, ILH and so on, and keep only the most commonly used acronyms in neurology/neuroscience = TMS, MEP, FDI, M1, and EEG.
6) The findings on TEPs are interesting but the Discussion section is rather thin. Can the authors talk more, for example, why do differences occur within the first 100 msec? Does the higher power represent overexcitability in the left (ipsilesional) M1? How does link back to the interhemispheric imbalance following a stroke? The authors may refer to some published articles e.g. Murase et al., 2004 (Cohen's group who is usually taken to be the first to report the imbalance in stroke); Dodd et al., 2017
7) Discussion: perhaps the authors can talk more on the centre ON-surround OFF theory? Even if the findings are intriguing, the authors may talk more with hypothetical explanations or with future work to continue this.
Minor:
1) Replace “both hemispheres (affected and unaffected)” to be “both affected and unaffected hemisphere” for readability in the Abstract.
2) The word sounds off in the Abstract: “Here we studied the intra-cortical excitability properties administering transcranial magnetic”
3) If the healthy volunteers and the stroke group are indeed age-matched (as mentioned in the Abstract), write this again in the Methods (Subjects) section.
4) To avoid confusion, it is better to clearly write in point 1 of Section 2.3.2 or as a caption in the relevant Figure 1-3, that the t = 0 ms is the onset of TMS stimulus, and all curves are aligned to this time point.
5) Section 3.2: “approximation to gaussian (assessed by Shapiro-Wilk” become “approximation to a Gaussian distribution, as assessed by ….”
6) Figure 4 and Figure 5: I recommend using the same scale for the Y-axis. This will ease readers in comparing the data visually.
Author Response

(The authors gave the same response as above.)

Round 2
Reviewer 3 Report
The authors have addressed the concerns. Congratulation.
Abstract Line 27: missing "of" before "delivering".